# Effectiveness of a 'Workshop on Decluttering and Organising' programme for teens and middle-aged adults with difficulty decluttering: a study protocol of an open-label, randomised, parallel-group, superiority trial in Japan

Yasuko Aso,[1,2] Kazue Yamaoka,[2] Asuka Nemoto,[2] Yuki Naganuma,[3] Masashige Saito[4]

► Additional material is published online only. To view please visit the journal online (http://dx.doi.org/10.1136/bmjopen-2016-014687)

For numbered affiliations see end of article.

**Correspondence to**
Dr Yasuko Aso;
asou@med.teikyo-u.ac.jp,
ya.poco.7.km@gmail.com

## ABSTRACT

**Introduction** Hoarding disorder can cause problems with work performance, personal hygiene, health and well-being. The disorder is a growing social problem in Japan. Having difficulty discarding rubbish, decluttering and organising can signal a future hoarding disorder, and early intervention is important. We developed an educational workshop on decluttering and organising for teens and adults with difficulty organising. The objective of this study is to evaluate the effectiveness of a workshop for reducing clutter and improving quality of life among younger people with difficulty decluttering and organising.

**Methods and analysis** An open-label, parallel-group, randomised controlled trial will be conducted among volunteers aged 12–55 years with mild difficulty decluttering and organising. Those in the intervention group will attend the workshop and receive a visit from a professional cleaning company to declutter their living space. The control group will have only the latter. The primary outcome will be the score on the Japanese version of the Saving Inventory-Revised. Secondary outcomes will be scores on the Clutter Image Rating Scale, the Japanese version of the Rosenberg Self-Esteem Scale and the Roles of Private Space Scale. The results will be examined for differences between the two groups in changes from baseline to 7 months. We will examine crude effects and adjust for gender and age using a general linear model for continuous variables and a logistic regression model for dichotomous variables. Sample size was calculated assuming a significance level of 5% (two tailed), a power of 80% and an effect size of 0.75. In total, 60 subjects (30 in each group) will be required.

**Ethics and dissemination** The study protocol has been approved by the Medical Ethical Committee of Teikyo University (No. 15-065). The findings will be disseminated widely through peer-reviewed publication and conference presentations.

### Strengths and limitations of this study

► The study will be an open-label, parallel-group, randomised controlled trial and, to our knowledge, is the first such study on this subject in Japan.
► The study will test a simple but feasible intervention for those with difficulty decluttering and organising.
► The outcomes measures are clearly set using reliable scales.
► The test is adequately powered to detect effects on the outcomes.
► Blinding is not possible because of the type of intervention, which may affect the results.

**Trial registration number** UMIN000020568. Issue date: 16 January 2016.

## INTRODUCTION

Having a hoarding disorder can cause problems with work performance, personal hygiene, health and well-being. Such disorders occasionally result in individuals living in environments with large amounts of garbage. Key symptoms of hoarding include difficulty discarding, excessive clutter and excessive acquisition of objects.[1 2] The latter symptoms pose a number of dangers, such as the potential for fire and tripping over clutter. The presence of trash or clutter in one's living space also affects an individual's overall quality of life.[3] This issue is a growing social problem in Japan.[4]

An individual living in such conditions can become isolated from the rest of society; this can be particularly true for older people.[5]

Unsanitary living conditions can eventually become a public health problem if they affect one's neighbours. Hoarding disorder is therefore an individual problem and a problem for society, and a practical solution is warranted from a public health point of view.

Hoarding disorder is often accompanied by psychiatric disorders such as attention deficit hyperactivity disorder (ADHD), depression or obsessive–compulsive disorder (OCD).[6–11] A relationship between hoarding disorder and distress tolerance and anxiety sensitivity has also been noted in a sample of young adults.[12] Signs of hoarding were separately reported in children aged 13[13] and 11–15 years,[14] and the severity of the disorder evidently increases with age. Many sufferers report symptom onset before age 20 years,[14] with symptoms tending to worsen past age 55 years.[13] This suggests that prevention among younger people is important. To accommodate this, we targeted a wide range of participants aged 12–55 years old.

A previous study in England and Germany estimated that incidence of hoarding disorder was 1.5%–5.8%.[15 16] Another study reported that cognitive–behavioural therapy, either individually or in groups, as well as home visits, are effective at assisting patients with hoarding disorder. However, these cognitive–behavioural therapies required relatively long intervention periods, and the subject samples were middle-aged and elderly people.[17–19]

To the contrary, a study on the effects of a brief anxiety sensitivity reduction intervention on obsessive–compulsive (OC) spectrum symptoms in a young adult sample showed reduced OC symptoms across the postintervention follow-up period. However, the intervention did not show a specific effect on reducing hoarding symptoms.[12] There is a clear need to identify new avenues through which to develop additional interventions.

Japan's population is ageing at an increasingly rapid rate. Given the association with age, early intervention is imperative. There is urgent need to establish a system to help prevent younger people who show signs of difficulty decluttering and organising from developing a hoarding disorder.

A previous study in Japan examined the state of the living space of 450 undergraduates from two universities.[20] The results showed that younger people who had difficulty decluttering and organising tended to live in cluttered or disorganised living spaces and to have low self-esteem. The study used the Japanese version of the Saving Inventory-Revised (SI-R).[21] This was developed in the USA[2] and translated into Japanese, and its reliability and validity have been verified.[2] The previous study's SI-R scores had a mean of 32.1 points (±13.0) and a normal distribution. SI-R scores have been reported to be high in people who said that they did not like to clean, but many such people also want to improve the state of their living space.[20] There are various ways to learn how to clean effectively in Japan, including workshops, books and professional support, but it is not clear which method is most effective.

We have therefore developed an education programme, of a series of workshops on decluttering and organising, for teens and adults who find it difficult to organise. In the workshops, we will help identify what belongings a person truly needs. It also covers the state of the individual's living space and its relationship with physical and mental health. The purpose of the workshop is to improve recognition of the importance of cleaning and to change behaviour.

This study examines behaviour 7 months after the intervention, using a control group, and is designed to assess both changes in habits and levels of stress. It is impossible to blind this type of intervention, so the study was made open label to minimise bias.

The aim of the study is to evaluate the effectiveness of the programme in reducing clutter and improving the quality of life in teens and middle-aged adults who have difficulty decluttering and organising and have a cluttered living space.

## METHODS AND ANALYSIS
### Study design
The design of this study is an open-label, parallel-group, stratified, randomised controlled trial. The aim of this trial is to investigate the effect of the workshop plus a professional organising and cleaning service. We hope that the programme will improve quality of life and help prevent hoarding disorder in younger people who currently have difficulty decluttering and organising. Participants will be involved for a period of 7 months. The design of the trial is shown in figure 1. Based on previous experience, it is expected that the study will include around 90% of eligible participants. Perioperative protocols are standardised.

### Participants
The study will look for volunteers aged between 12 and 55 years and who have mild organisation difficulties. Volunteers will need to meet all inclusion criteria below and not meet any of the exclusion criteria (see box 1).

### Recruitment and allocation
#### Subject recruitment
We will recruit participants in several ways, including through posters put up in visiting nursing stations, visiting care stations, community pharmacies and at a university, high school and junior high school in northwest Tokyo. We will also recruit online using a homepage, blog and social media for an events corporation, an association of organisers and a non-profit organisation that provides support for people with ADHD.

### Subject assignment
Applicants will visit Teikyo University to receive an explanation of the study objectives. After assessment via questionnaire, those verified as eligible will be immediately enrolled if they give their full consent. Each subject will then be randomly assigned to either the

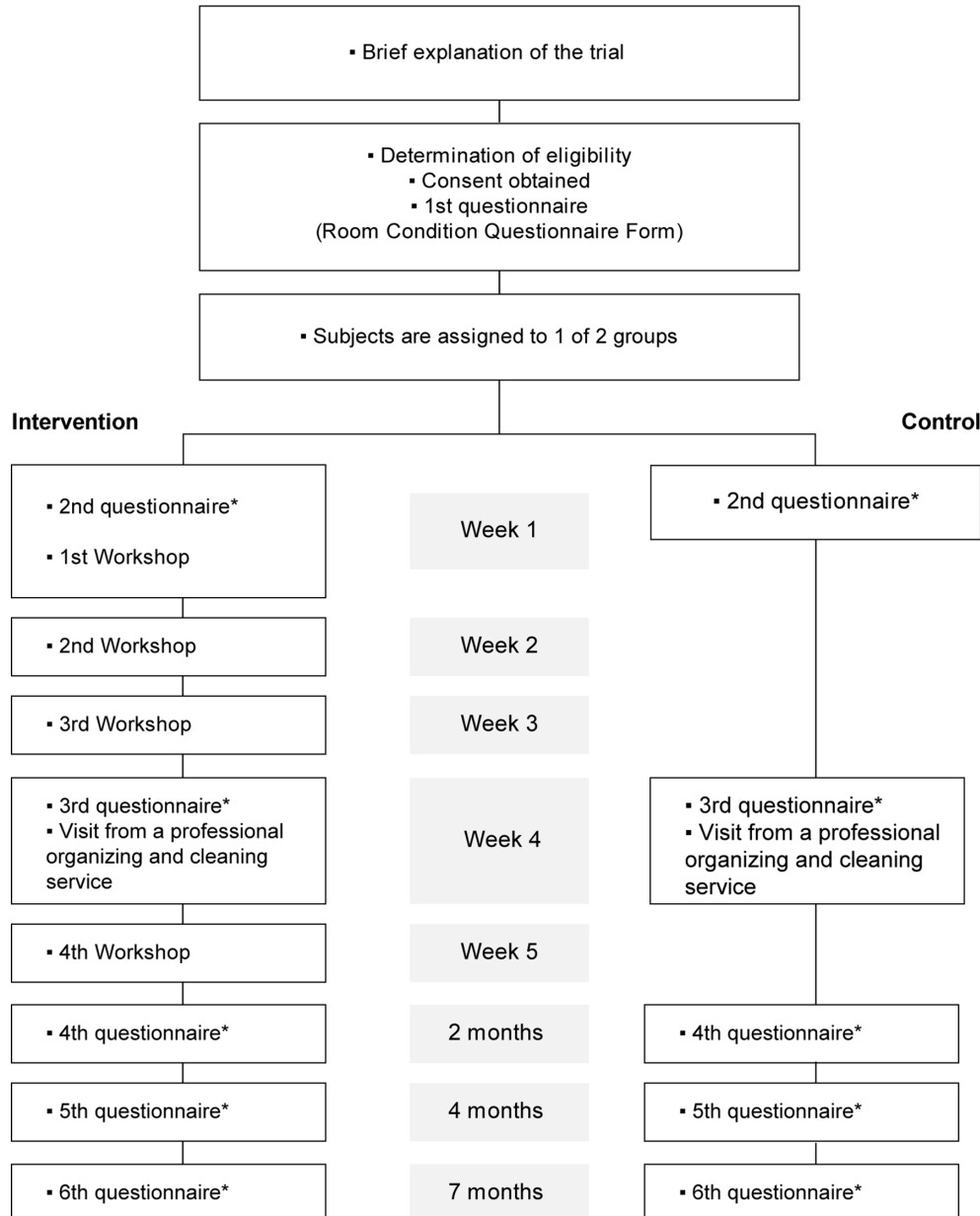

* Room Condition Questionnaire Form. The participants will supply photos of their room/house with their completed questionnaire.

**Figure 1** Trial design.

intervention or control group. Enrolment will continue until the required sample size (30 participants per group) is reached. We consider that the effects of lifestyle education need several months to show progress; many studies use 6 months or longer.[22 23] We therefore set a term of 7 months by considering 6 months after the intervention begins as the observation period in the present study.

### Randomisation and blinding

A sealed envelope method will be used for random allocation. This is based on a permuted block method, with a block size of 4 and a random number table created before the study begins. Each presealed envelope, prepared by a person independent from the research,

will contain a random number. Participants will open this at the location of the session meeting where they gave consent and cannot change their designated group.

Because of the open-label design, participants will know whether they are assigned to the intervention or control group. However, this information is blinded for the professional organising and cleaning service in order to avoid risk of bias.

Effectiveness of the intervention will be determined via surveys completed by the participants and sent directly to the study chair at Teikyo University along with a photograph of the room to assess the present level of cleanliness or clutter. The organising and cleaning services will not be able to see the results for individual subjects.

## Box 1　Inclusion criteria and exclusion criteria

*Inclusion criteria*
- ► Individuals who have difficulty decluttering and organising. (SI-R score≥30).
- ► Individuals with a total score of four points or more on the Japanese version of the Clutter Image Rating (CIR) scale*.
- ► Individuals who are responsible for decluttering and organising their own rooms.
- ► Individuals with no planned relocations, extensive renovations of living space or room changes (to allow comparison of the state of living spaces before and after the intervention).
- ► Individuals who are able to attend at least three workshop sessions and receive visits from an organising and cleaning service.
- ► Individuals who responded in the questionnaire, 'I want to organise my room according to my will'.
- ► Individuals who can send photographs of their living spaces at the specified time, that is, without delay.
- ► Individuals or a roommate/family member (the informed consent form for minors if the individual is under the age of 15 years) who consent in writing to their participation in the study and to having a better organised and tidied living space. If the individual is a minor, consent must be provided by a guardian.
- ► Individuals between the ages of 12 and 55 years at the start of the study.

*Exclusion criteria*
- ► Individuals who are unable to organise and tidy up their living spaces during the study period, for example:
  - ► individuals with an illness or disability
  - ► individuals who are not living at the registered residence or are physically unable to regularly organise and tidy up.
- ► Individuals living in a space that is too messy (an organisation expert and two cleaning experts judge the space and deem whether it can be sufficiently organised within about 3 hours).
- ► Individuals who harass other attendees while attending the workshop or who are deemed incapable of obeying the specified rules of attendance.
- ► Individuals who are rendered incapable of participation during this study.
- ► Individuals suffering from a condition or disability that is deemed to preclude study participation. If there are difficulties with that determination, an individual may be barred from participation at the discretion of a physician.
- ► Individuals who are otherwise deemed ineligible by an investigator (or subinvestigator).

*The CIR Scale score for the general public is a mean of 3.8 points (±1)[20] so the subjects in this study must have at least that score.
SI-R, Saving Inventory-Revised.

## Interventions

### Intervention group

This group will attend a workshop programme, undergo inspections of cleanliness or clutter of their rooms and receive visits from an organising and cleaning service. The workshop will consist of four sessions. Room cleanliness or clutter will be inspected six times. The first inspection will take place once eligibility is verified and consent is obtained. The second will take place before the start of the study. Others will take place 4 weeks, 2 months, 4 months and 7 months after the start of the study (see figure 1).

### Control group

The control group will receive an initial inspection to evaluate room cleanliness or clutter once eligibility is verified and consent obtained. The second inspection will take place before the intervention group starts the workshop. The control group will receive a visit from a professional cleaning company only to declutter the living spaces. Further inspections will take place 4 weeks, 2 months, 4 months and 7 months after the start of the study. Members of the control group who wish to attend the workshop may do so after the study period ends.

Additionally, until the end of the research period 7 months after commencement of the study, no participants are allowed to directly request any cleaning service be done.

### The content of the workshop

The workshop's purpose is to improve participant behaviour and develop habits of decluttering and organising. The workshop starts by covering the relationship between physical and mental health and the state of the living space and then discusses the participants' ideal life. A study representative has developed a workshop programme based on material from the Japan Association of Life Organizers and Edison Club, a non-profit organisation supporting patients with ADHD, their families and teachers. The textbook used in the workshop is based on a text used in elementary schools[24]; therefore, 12-year-olds should have no difficulty understanding the contents.

A public health nurse and an organiser will deliver four lectures, with the former leading the first 30 min of each lecture and the latter the final 1.5 hours (see table 1 for details).

### Outcome measures and subject's characteristics

#### Primary outcome

The primary outcome is the score on the Japanese version of the SI-R. The SI-R score is an index used to gauge the difficulties posed by a hoarding disorder (a higher score indicates a greater tendency to hoard). It measures mental anguish caused by hoarding and the impacts of hoarding on life. A higher score indicates a greater tendency to hoard.[2 21] There is presently no Japanese version of the Child Saving Inventory.[25] The academic ability of Japanese teenagers, however, ranks among the top in the world.[26] Only 292 Japanese *kanji* characters are used for the Japanese version of the SI-R, and most have already been learnt in elementary school. Therefore, the Japanese version of the SI-R is applicable for junior high school students and should not create problems related to comprehension.

The primary effect size is the difference changes in SI-R score from baseline to 7 months between the two groups. The hypothesis of this study is that the mean SI-R score of the intervention group will decrease more than that of the control group.

**Table 1** Syllabus for 'Workshop on Decluttering and Organising'

| | |
|---|---|
| 1st (2_hours) | <Reduce><br>► Orientation and goal setting (public health nurse).<br>► Understanding the importance of working as a group (public health nurse).<br>► Three fundamentals for healthy and comfortable living (reduce, organise and maintain) (organiser).<br>► Organising in small steps: around your desk and your bed (organiser).<br>► Explanation of homework (public health nurse). |
| 2nd (2_hours) | <Organise><br>► Review of previous session (public health nurse).<br>► The relation between decluttering and organising and health (fall prevention and allergic disease) (public health nurse).<br>► Organising in small steps: the kitchen (organiser).<br>► Explanation of homework (public health nurse). |
| 3rd (2_hours) | <Review><br>► Review of previous session (public health nurse).<br>► Solutions for people who have difficulty discarding things and buy (or receive) unnecessary things (public health nurse).<br>► Organising in small steps: a place to relax (organiser).<br>► Explanation of homework (public health nurse). |
| 4th (2_hours) | <Maintain><br>► Review of previous session (public health nurse).<br>► Organising in small steps: closets and wardrobes (organiser).<br>► Schedule of questionnaire submission (public health nurse). |

## Secondary outcomes

Secondary outcomes include:

► Score on the Clutter Image Rating (CIR) Scale, which measures clutter using photographs of rooms (a higher score indicates more clutter).[27 28]

► Score on the Japanese version of the Rosenberg Self-Esteem Scale.[29] This measures the degree of self-respect and the value placed on the self. A high score shows stronger feelings of self-esteem.

► Score on the Roles of Private Space Scale. This linear measure observes the function, the necessity and securement level of private space.[30]

These outcomes will be used to examine the amount of decluttering and organising, including frequency of cleaning and number of visitors in a week. They will allow the researchers to assess the state of subjects' living spaces and their ability to declutter and organise because of the workshop.

## Subject characteristics

During the first meeting, participants will provide information about their gender, age, comorbidities, previous history, medication, number of roommates/family members in their household, number and size of living spaces, how often their living space is organised and tidied up, number of visitors to their living space in the previous week, whether they have difficulty decluttering and organising and are living with a roommate or family members and aspects of social status such as occupation, marital status, educational background and work pattern (whether they are working on a shift basis).

## Statistical analysis

### Sample size

A previous study divided 46 patients with a hoarding disorder (mean age: 53.9 years) into two groups. One group of 23 subjects was scheduled for 26 sessions of cognitive–behavioural therapy approximately on a weekly basis. The second group of 23 subjects had yet to undergo cognitive–behavioural therapy.[19] A comparison of the two groups indicated that subjects who had completed the 12th session of cognitive–behavioural therapy had a 10-point drop in their SI-R score. Around 10 points were improved on using a pre-test on the general population. A meta-analysis in another study indicated that intervention was more effective in younger individuals than older ones. This study therefore assumed a potential 10-point improvement in SI-R score.[19] Previous results indicated that the SI-R score had a normal distribution and SD was 13.0.[20] The sample size was therefore calculated as 28 per group under the assumptions of a significance level of 5% (two tailed), a power of 80% and an effect size of 0.75. Considering the likely dropout rate, we set the sample size needed for this trial as 60 subjects in total, 30 in each group.

### Statistical analysis

Analyses will be conducted under the intention-to-treat principle. The main target set for analysis is the full analysis set. Missing data will be permutated under the assumption of missing at random following the last observation carried forward principle. A sensitivity analysis will be performed using the multiple imputation method.

The secondary target set for the analyses will be the per-protocol set.

Summary statistics (maximum, median, minimum, mean and SD) will be calculated for all continuous data, and frequency and proportion for categorical data.

The differences in effect sizes between the two groups will be examined using a t-test. Differences will be assessed in a general linear model by adjusting for gender, age and marital status at the start of the study and outcome measure at the baseline. Crude and adjusted ORs will be calculated for binary variables and the two groups will be compared using logistic regression analysis. As secondary analysis, mixed-effects random-effects models will be used as a longitudinal data analysis. The significance level for testing will be set at $p < 0.05$ (two tailed).

### Timeframe of the study
Participants were registered between January and May 2016. Eight workshops have been run between 3 February and 11 June 2016. Data are currently being collected.

### Data management
Personal information obtained in this study will be coded. Participant files will be stored in numerical order in a secure but accessible place and manner. Data will be anonymised before analysis. Data will be kept on a password-protected computer with limited access.

The study organiser will keep all documents related to this trial in a locked cabinet. Five years after the study is presented, or 3 years after the final publication of the results (whichever is later), all documents will be disposed of using a document destruction service contracted by the University.

### Conditions for discontinuation of this study and actions in that event
When significant information about the safety or effectiveness of the intervention is obtained, the study organiser will determine whether to continue the study.

### Treatment of subjects after this study is conducted
If the intervention group is performing significantly better than the control group at the end of the study, the control group will be informed. Members of the control group who wish to do so may then attend the workshop after the conclusion of this study.

### Monitoring
Information on when the study begins, the conduct of the study (sample size), ethical considerations that have been made, the occurrence of any detrimental or adverse events, the results of the study and registration of the study with a public database will be submitted to the ethics committee in an annual interim report. A report will also be submitted to the ethics committee on the conclusion of the study and when the final results are presented.

### Protocol amendments
If any occur, the ethics committee may be notified as necessary.

### Follow-up of adverse events
This study involves attending a workshop programme to help people declutter and organise and receiving visits from a professional organising and cleaning service. There is little likelihood of any health hazards to the subjects. If any serious adverse events occur, they will be reported in line with the standard operating procedure on Reporting Serious Adverse Events in Clinical Research.

### Ethics and dissemination
The Medical Ethical Committee of Teikyo University (No. 15-065) has approved the study protocol. The findings will be disseminated widely through peer-reviewed publication and conference presentations.

Study participant candidates will be given an explanation form. The study will be fully explained orally and in writing, and voluntary consent for study participation will be sought in writing. The explanation and consent form is shown in online supplementary files 1–8. Considerations have been made in the event a participant consents to take part but later withdraws that consent. Participants will be assured they will not be penalised for withdrawal or for not participating. Participants will be informed of the study's results after its conclusion and on their request. Those aged 12–14 years must provide written informed assent together with an adult consent form from the parents. Those aged 15–18 years must provide written consent (same as for adults) and a guardian consent form. The ages for requisite informed assent were decided with reference to the guidelines of the American Academy of Pediatrics[31 32] and the decision of the Teikyo University Ethics Committee. Minors living with a guardian or family member (in this section hereinafter 'relevant party') will have a similar form mailed to the relevant party or the minor will be asked to deliver it to the relevant party. If information is obtained that may affect the participant or the relevant party, or if the protocol is modified to such an extent that the change would affect the consent of all relevant parties, that information will be promptly provided to the relevant parties and the participant's desire to continue will be verified. In either of the aforementioned instances, the Research Ethics Committee will revise and approve the consent and explanation form, and the consent of all relevant parties will be sought.

Participants who wish to view their information or withdraw consent may contact the study chair for resolution. Any modifications to the protocol that may affect the conduct of the study will be presented to the aforementioned committee.

### DISCUSSION
One aim of this study is to assess the effect of an intervention (a workshop) on the state of living spaces and the

quality of life of younger people who have difficulty decluttering and organising and who therefore have a cluttered living space. If the approach is found to be effective, then the findings of this study will be widely publicised and the workshop materials will be made widely available, so that it can be conducted by others, including administrative bodies and professional organisers and cleaners. This should improve the quality of life and help to prevent a hoarding disorder in younger people who have difficulty decluttering and organising.

This study has certain strengths and weaknesses. Blinding is not possible because of the type of intervention. It is impossible to create a convincing placebo for the workshop on decluttering and organising, and this lack of blinding may affect the results. As far as possible, subjects need to have an experience equivalent to undergoing training from a professional organising and cleaning service.

**Author affiliations**
[1]Department of Nursing, Faculty of Health Sciences, Tsukuba International University, Tuchiura, Ibaraki, Japan
[2]Teikyo University Graduate School of Public Health, Tokyo, Japan
[3]Department of Nursing, Faculty of Nursing, Tokyo Ariake University of Medical and Health Sciences, Tokyo, Japan
[4]Department of Social Welfare, Faculty of Social Welfare, Nihon Fukushi University, Aichi, Japan

**Acknowledgements** We are grateful for advice from Yukiko Mochizuki, Yuka Nojiri and Mihoko Shimozono about preparing a study protocol. The text for the workshop was prepared with help from Keiko Takayama and Mayumi Takahara.

**Contributors** YA conceived the study. KY and AN participated in the development of the protocol. MS and YN contributed to the study conception and design of the education programme. All authors prepared and revised the manuscript, including relevant scientific content. All authors approved the final version of the manuscript.

**Funding** This work was supported by JSPS KAKENHI Grant Number JP26671045 (2014–2017) Grant-in-Aid for Scientific Research on 'The state of young people who chronically fail to organize or tidy up and development of a program for effective intervention' (Challenging Exploratory Research, Study Organizer: YA, grant no. 26671045).

**Disclaimer** This funding source had no role in the design of this study and will have no role during its execution, analyses, interpretation of the data or in any decision to submit results.

**Competing interests** None declared.

**Patient consent** Obtained.

**Ethics approval** Medical Ethical Committee of Teikyo University (No. 15-065).

**Provenance and peer review** Not commissioned; externally peer reviewed.

**Data sharing statement** Exhibit 7: Explanation Document (themselves from 15 years old or older and parents and those who live together). Exhibit 8: Research Explanation Document 'From 12 years old to 14 years old' (informed assent form for children). Exhibit 9: Consent Form (subjects themselves from 15 years old or older and parents and those who live together). Exhibit 10: Consent Form (subject themselves from 12 years old to 14 years old).

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
