## [Reviewer comments · BMJ Open]

ARTICLE DETAILS

TITLE (PROVISIONAL)	Effectiveness of a “Workshop on Decluttering and Organizing” program for teens and middle-aged adults with difficulty decluttering: a study protocol of an open-label, randomized, parallel-group, superiority trial in Japan
AUTHORS	Aso, Yasuko; Yamaoka, Kazue; Nemoto, Asuka; Naganuma, Yuki; Saito, Masashige

VERSION 1 - REVIEW

REVIEWER	Volen Ivanov Karolinska Institutet, Department of Clinical Neuroscience, Sweden.
REVIEW RETURNED	07-Nov-2016

GENERAL COMMENTS	This study focuses on delivering and evaluating the effects of a workshop on decluttering and organizing in the home. People with hoarding disorder can often be reluctant to seek help for their hoarding difficulties and those who do seek treatment do this many years after symptom onset. This pattern probably contributes to the difficulties associated with the treatment of hoarding. The study protocol describes a prevention approach to hoarding disorder that might potentially have an important effect on the development of the disorder in individuals at risk and might provide an important contribution to the treatment/prevention of hoarding once conducted. In sum, I find the study protocol to be suitable for publication. I would like to suggest a few possible improvements that the authors might take into consideration. - The introduction might benefit from additional references to relevant work in the area that could strengthen the rationale for using the specified intervention. Studies such as “Effects of a brief anxiety sensitivity reduction intervention on obsessive compulsive spectrum symptoms in a young adult sample” by Timpano et al and references on “The buried in treasures –workshops” might be relevant here. Also, given that the authors aim to treat a younger population I would suggest providing some more background on hoarding in children and adolescents.- I find the inclusion and exclusion criteria to be a bit vague. For example the criterion: “Individuals who want to have a more organized and tidy living space”. Could this criterion be operationalized in some way? Another problematic criterion is: “Individuals in a living space that is so highly cluttered that they are unable to organize and tidy up that space themselves.”. Here the authors might include a cut off on the Clutter Image Rating for example.- My main concern with the intervention described is that it aims to target difficulties in both teens and middle aged adults. I find it difficult to understand that the same intervention could be delivered
---

	to someone aged 12 and someone who is 50 years old without considerable adjustments in the workshop content and delivery. It is also not known whether teens and adults who have difficulties organizing etc. are actually displaying the same problem.
--	---

REVIEWER	Christie Burton Hospital for Sick Children Canada
REVIEW RETURNED	14-Nov-2016

GENERAL COMMENTS	Aso and colleagues present a study protocol for a randomized control trial that tests the effectiveness of a workshop on decluttering and organizing on hoarding symptoms. First I want to commend the authors on developing some strategies to try to prevent hoarding disorder and targeting younger individuals for this trial. Overall the protocol is well written and clear. Some additional information about the workshop, clarification about group assignment, and a bit more rationale about some of the study parameters would be helpful. Also, the authors may want to consider an alternative measure for younger participants and methods of analyzing group differences. Introduction 1. The authors mention co-morbid disorders with hoarding such as depression and ADHD but it's unclear why OCD is not mentioned along with them. 2. In the introduction the authors state "the workshop emphasizes the importance of having a sense of values in life". This statement is a bit unclear – what will be taught about values specifically? Methods 3. Because appendices were not available, it's not clear the exact content of the workshops and these details would be helpful. It would also be good to know how long each workshop session lasts (e.g., 2 hours, etc). How was the workshop developed and by whom? 4. Why was the age range of 12-55 chosen? 5. Why was the 7-month interval chosen? Would it be possible to include another follow-up session a little further out since the gains may or may not last. 6. Under Subject Recruitment, the authors discuss the results of a previous study showing the distribution of comorbidities such as depression and ADHD in the hoarding group and then the authors state " participants will be recruited in several ways so that the study population is reasonably representative". Does this mean that the authors will try to get the same distribution of comorbidities in their sample? Is this even necessary? 7. The SI-R was developed and validated in a sample that was at youngest 18. There is a child-friendly version of the SIR – the Child Saving Inventory (CSI) developed by Eric Storch. The authors may want to consider using this for their participants less than 18. 8. Is examining the difference in effect size the optimal way to examine group differences of the pre-vs.-post effect of the intervention? Using a repeated measures model would allow you to examine any differences at baseline and provide more information about any changes that might occur after the intervention. 9. Sample Size: Basing the estimated score improvement from a CBT study may over optimistic. CBT studies would likely have participants who are more impaired (and thus have more room for
--

	improvement) and the interventions may be more involved. This should be considered when estimated sample sizes and power. 10. In the study protocol section (pg. 2) it states “During enrollment subjects will be randomly chosen to attend or not attend a Workshop... based on whether or not they have been trained at organizing and tidying up”. Does this mean that assignment to the intervention or control group is dependent on previous experience in organizing and cleaning? This sentence seems at odds with the rest of the protocol and random assignment that was described. Please clarify. Minor comments:  - In abstract, it's not clear why hoarding disorder is pluralized.
--	--

VERSION 1 – AUTHOR RESPONSE

Dear Dr. Volen Ivanov (Reviewer 1):

This study focuses on delivering and evaluating the effects of a workshop on decluttering and organizing in the home. People with hoarding disorder can often be reluctant to seek help for their hoarding difficulties and those who do seek treatment do this many years after symptom onset. This pattern probably contributes to the difficulties associated with the treatment of hoarding. The study protocol describes a prevention approach to hoarding disorder that might potentially have an important effect on the development of the disorder in individuals at risk and might provide an important contribution to the treatment/prevention of hoarding once conducted. In sum, I find the study protocol to be suitable for publication. I would like to suggest a few possible improvements that the authors might take into consideration.

Thank you for your valuable comments. We have added more detailed descriptions in each section following the reviewers’ suggestions. The revised sentences and additions are shown in red text. We hope that the responses are clearly presented. We appreciate your careful review.

- The introduction might benefit from additional references to relevant work in the area that could strengthen the rationale for using the specified intervention. Studies such as “Effects of a brief anxiety sensitivity reduction intervention on obsessive compulsive spectrum symptoms in a young adult sample” by Timpano et al and references on “The buried in treasures –workshops” might be relevant here. Also, given that the authors aim to treat a younger population I would suggest providing some more background on hoarding in children and adolescents.

We added the reference and added the sentence “Its relationship with distress tolerance and anxiety sensitivity has also been pointed out in a sample of young adults. [11]” Thank you for your kind suggestion.

We also provided some more background on hoarding in children and adolescents as “These points suggest prevention from the young generation is important and we therefore targeted the ages of 12–55 years old. As age 12 is the middle junior high school entrance age in Japanese education system, we used this instead of 13.”

- I find the inclusion and exclusion criteria to be a bit vague. For example the criterion: “Individuals who want to have a more organized and tidy living space”. Could this criterion be operationalized in some way? Another problematic criterion is: “Individuals in a living space that is so highly cluttered that they are unable to organize and tidy up that space themselves.”. Here the authors might include a

cut off on the Clutter Image Rating for example.

We corrected the descriptions in Table 1 as follows.

- Individuals who responded in the questionnaire, “I want to organize my room according to my will”.
- Individuals living in a space that is too messy (state of clutter judged as too great to organize even in a 2.5-hour visit by two experts).

- My main concern with the intervention described is that it aims to target difficulties in both teens and middle aged adults. I find it difficult to understand that the same intervention could be delivered to someone aged 12 and someone who is 50 years old without considerable adjustments in the workshop content and delivery. It is also not known whether teens and adults who have difficulties organizing etc. are actually displaying the same problem.

We think that our method of workshop is very simple and even among younger age groups (such as junior high school students) can perform well.

Dear Dr. Christie Burton (Reviewer 2):

Aso and colleagues present a study protocol for a randomized control trial that tests the effectiveness of a workshop on decluttering and organizing on hoarding symptoms. First I want to commend the authors on developing some strategies to try to prevent hoarding disorder and targeting younger individuals for this trial. Overall the protocol is well written and clear. Some additional information about the workshop, clarification about group assignment, and a bit more rationale about some of the study parameters would be helpful. Also, the authors may want to consider an alternative measure for younger participants and methods of analyzing group differences.

Thank you for your valuable comments. We have added more detailed descriptions in each section following the reviewers' suggestions. The revised sentences and additions are shown in red text. We hope that the responses are clearly presented. We appreciate your careful review.

Introduction

1. The authors mention co-morbid disorders with hoarding such as depression and ADHD but it's unclear why OCD is not mentioned along with them.

We added OCD. Thank you for your suggestion.

2. In the introduction the authors state “the workshop emphasizes the importance of having a sense of values in life”. This statement is a bit unclear – what will be taught about values specifically?

We corrected the explanation for this as, “In the workshops, we emphasize importance of the sense of values (such as necessity) that each person has for things.”

Methods

3. Because appendices were not available, it's not clear the exact content of the workshops and these details would be helpful. It would also be good to know how long each workshop session lasts (e.g., 2 hours, etc). How was the workshop developed and by whom?

Thank you for your comments. We added and corrected Table 2 and related explanations in accordance with your comments.

4. Why was the age range of 12-55 chosen?

We considered that prevention in the younger generation is important and therefore targeted ages 12–55 years. Age 12 is the middle junior high school entrance age in the Japanese education system, so we used that age instead of 13.

We also provided some more background on hoarding in children and adolescents in the Introduction section.

5. Why was the 7-month interval chosen? Would it be possible to include another follow-up session a little further out since the gains may or may not last.

We considered that the effect of lifestyle education needs several months to yield results, and many studies adopted 6 months or more [22-23]. We therefore set 7 months by considering 6 months after the beginning of the intervention as the observation period in this study.

6. Under Subject Recruitment, the authors discuss the results of a previous study showing the distribution of comorbidities such as depression and ADHD in the hoarding group and then the authors state “ participants will be recruited in several ways so that the study population is reasonably representative”. Does this mean that the authors will try to get the same distribution of comorbidities in their sample? Is this even necessary?

No, that is not the intent. We corrected the sentence as, “We considered recruitment so the participants would be representative of a society that includes hoarding behavior.”

7. The SI-R was developed and validated in a sample that was at youngest 18. There is a child-friendly version of the SIR – the Child Saving Inventory (CSI) developed by Eric Storch. The authors may want to consider using this for their participants less than 18.

Thank you for your advice. Regarding the SI-R questionnaire, however, we do not have a Japanese version of the CSI and we felt we could use the SI-R for adolescents under age 12 by the pre-test. We therefore used SI-R in this study. We added explanation for this.

8. Is examining the difference in effect size the optimal way to examine group differences of the pre-vs.-post effect of the intervention? Using a repeated measures model would allow you to examine any differences at baseline and provide more information about any changes that might occur after the intervention.

We will use repeated measures models as a secondary analysis. We added this in the statistical session as, “As secondary analysis, mixed-effects random-effects models will be used as a longitudinal data analysis.”

9. Sample Size: Basing the estimated score improvement from a CBT study may over optimistic. CBT studies would likely have participants who are more impaired (and thus have more room for

improvement) and the interventions may be more involved. This should be considered when estimated sample sizes and power.

The sample size calculation is based on our pre-test. We added this in the sample size section as, "Around 10 points were improved upon using a pre-test on the general population." Thank you for your comment.

10. In the study protocol section (pg. 2) it states "During enrollment subjects will be randomly chosen to attend or not attend a Workshop... based on whether or not they have been trained at organizing and tidying up". Does this mean that assignment to the intervention or control group is dependent on previous experience in organizing and cleaning? This sentence seems at odds with the rest of the protocol and random assignment that was described. Please clarify.

No, the assignment to the intervention or control group is independent of previous experience in organizing and cleaning. We added this in the randomization and blinding section.

Minor comments:

- In abstract, it's not clear why hoarding disorder is pluralized.

Thank you for your comment. We corrected it accordingly.

VERSION 2 – REVIEW

REVIEWER	Volen Ivanov Karolinska Institutet, Sweden
REVIEW RETURNED	20-Jan-2017

GENERAL COMMENTS	I think the authors have done a good job of addressing the previous reviewer comments and I would thus recommend accepting the manuscript for publication.
--

REVIEWER	Christie Burton Hospital for Sick Children, Canada
REVIEW RETURNED	10-Jan-2017

GENERAL COMMENTS	Upon re-review of the manuscript there are still several items that require clarification. Generally, the manuscript could be clearer, use more evidence to support statements and be more precise in language. Specifically, some of the information that was asked to be added to the manuscript is not always well integrated into the text and could be revised to improve the readability of the manuscript. 1. In the intro, the authors say that "HD occasionally takes the form of a house full of rubbish, where individuals accumulate trash or clutter over a prolonged period of time". Clutter is a diagnostic criterion and is only allowed to not be present if being controlled by a third party. This statement makes it sound like it only happens sometimes. It would also be good to mention the other characteristics of HD 2. In the intro, the authors state the age of onset of hoarding is from
---

	age 13 which is based on one reference but there are other studies showing that it can be earlier and that the median age of onset is 11-15 (Tolin et al., 2010). Please amend this sentence to represent the literature. 3. In the intro, paragraph 5, the authors added a sentence about the intervention study focused on anxiety sensitivity in response to a reviewers comment but it requires a bit of context so that it fits in with the paragraph. At the moment it feels as though it does not fit. 4. In the same paragraph the authors say why they used aged 13 but this sentence is out of place since the study has not been described or introduced yet. This should be moved elsewhere. 5. For the exclusion criteria, how to experts judge if a space will take longer than 2.5 hours to be organized? Is there an operationalized cut-off for this? 6. In response to the comment about concerns about using the same workshop for 12 and 55 year olds, the author responded that they think that younger age groups will perform well. Is there any evidence to support that statement? 7. In the Introduction, the authors revised their explanation of the emphasis of the workshop to say “We emphasize importance of the sense of values (such as necessity) that each person has for things”. This sentence assumes that the issue with clutter and disorganization is based on a lack of these values. Is this supported by any data? Is this the critical focus of the workshops? The description of the workshop in this section could be more precise and specific. 8. In the Subject Recruitment section, the authors added the sentence “We considered recruitment so that the participants would be representative of a society that includes hoarding behaviour” in response to a reviewer comment. This sentence is unclear and given that it follows the sentence talking about the distribution of hoarding and depression, it makes it seem as though this is the representative society which the authors say is not the intent. This needs to be clarified and perhaps the first sentence of this paragraph removed? 9. Intervention section: 1) Intervention group section: Who does the inspections and 2) control group section: does this group also receive the organizing cleaning service? It is stated that they do in the abstract. 10. The authors say that in a pre-test that it seemed reasonable to use the SI-R in people under 18 years of age – is there any data that they can very briefly summarize in the paper to support that?
--	--

VERSION 2 – AUTHOR RESPONSE

Reviewer: 1

Reviewer Name: Volen Ivanov

Institution and Country: Karolinska Institutet, Sweden

Competing Interests: None declared

I think the authors have done a good job of addressing the previous reviewer comments and I would thus recommend accepting the manuscript for publication.

Thank you very much for your kind consideration.

Reviewer: 2

Reviewer Name: Christie Burton
Institution and Country: Hospital for Sick Children, Canada
Competing Interests: None

Upon re-review of the manuscript there are still several items that require clarification. Generally, the manuscript could be clearer, use more evidence to support statements and be more precise in language. Specifically, some of the information that was asked to be added to the manuscript is not always well integrated into the text and could be revised to improve the readability of the manuscript.

Thank you for your valuable comments. We have added more detailed descriptions in each section following the reviewers' suggestions. The revised sentences and additions are shown in red text. We hope that the responses are clearly presented. We appreciate your careful review.

1. In the intro, the authors say that "HD occasionally takes the form of a house full of rubbish, where individuals accumulate trash or clutter over a prolonged period of time". Clutter is a diagnostic criterion and is only allowed to not be present if being controlled by a third party. This statement makes it sound like it only happens sometimes. It would also be good to mention the other characteristics of HD

Thank you for your comment. Accordingly, we added other keys symptom of hoarding in HD, such as difficulty discarding and excessive acquisition. (p. 3, line 3)

2. In the intro, the author's state the age of onset of hoarding is from age 13 which is based on one reference but there are other studies showing that it can be earlier and that the median age of onset is 11-15 (Tolin et al., 2010). Please amend this sentence to represent the literature.

Thank you for your comment. Accordingly, we changed the age and added the reference and corrected the sentence. (p. 3, lines 15–19)

3. In the intro, paragraph 5, the authors added a sentence about the intervention study focused on anxiety sensitivity in response to a reviewers comment but it requires a bit of context so that it fits in with the paragraph. At the moment it feels as though it does not fit.

Thank you for your advice. We changed the corresponding sentence. (p. 3, lines 27–36)

4. In the same paragraph the authors say why they used aged 13 but this sentence is out of place since the study has not been described or introduced yet. This should be moved elsewhere.

Thank you for your comment. We removed this because it is written on p. 4, line 31.

5. For the exclusion criteria, how to experts judge if a space will take longer than 2.5 hours to be organized? Is there an operationalized cut-off for this?

To clarify this, we added the following: "An organization expert and two cleaning experts judge the space and deem whether it can be sufficiently organized within about 3 hours." (p. 5, Table 1 Exclusion criteria)

6. In response to the comment about concerns about using the same workshop for 12 and 55 year olds, the author responded that they think that younger age groups will perform well. Is there any evidence to support that statement?

Thank you for your comment. To clarify this, we added the following: “The textbook used in the workshop is based on a text used in elementary schools [22]; therefore, 12-year-olds should have no difficulty understanding the contents.” (p. 6, line 39)

7. In the Introduction, the authors revised their explanation of the emphasis of the workshop to say “We emphasize importance of the sense of values (such as necessity) that each person has for things”. This sentence assumes that the issue with clutter and disorganization is based on a lack of these values. Is this supported by any data? Is this the critical focus of the workshops? The description of the workshop in this section could be more precise and specific.

Thank you for your comment. This is not a critical focus of the workshop. We have provided a detailed explanation of the workshop. (p. 4, line 6)

8. In the Subject Recruitment section, the authors added the sentence “We considered recruitment so that the participants would be representative of a society that includes hoarding behavior” in response to a reviewer comment. This sentence is unclear and given that it follows the sentence talking about the distribution of hoarding and depression, it makes it seem as though this is the representative society which the authors say is not the intent. This needs to be clarified and perhaps the first sentence of this paragraph removed?

Thank you for your suggestion. We removed the sentence.

9. Intervention section: 1) Intervention group section: Who does the inspections and 2) control group section: does this group also receive the organizing cleaning service? It is stated that they do in the abstract.

Thank you for your comments. Our responses are as follows.

- 1) The participants were given self-administered questionnaires.
- 2) The control group will only receive a visit from a professional cleaning company to declutter the living space. We added this explanation in the manuscript. (p. 6, lines 15-16, 25-26)

10. The authors say that in a pre-test that it seemed reasonable to use the SI-R in people under 18 years of age – is there any data that they can very briefly summarize in the paper to support that?

To clarify this, we added the following sentence. (p. 7, line 10-15)

“The academic ability of Japanese teenagers, however, ranks among the top in the world. [24] Only 292 Japanese kanji characters are used for the Japanese version of the SI-R. And most of these kanji have already been learned in elementary school. In a pilot study on elementary school sixth graders, no comprehension problems were encountered.” (Unpublished)

VERSION 3 – REVIEW

REVIEWER	Christie Burton Hospital for Sick Children, Canada
REVIEW RETURNED	03-Mar-2017

GENERAL COMMENTS	The revised manuscript takes into account the previous author concerns. In order for this paper to be suitable for publication, it needs to be proof-read and edited thoroughly. There are typos, grammatical errors and unclear sentences.
---

VERSION 3 – AUTHOR RESPONSE

Responses to the comments of Reviewer #2

1. The revised manuscript takes into account the previous author concerns. In order for this paper to be suitable for publication, it needs to be proof-read and edited thoroughly. There are typos, grammatical errors and unclear sentences.

Response: Thank you very much for your advice. We have confirmed the contents and used a professional English editing service. We will also submit a certificate as verification of this editing.